# Association between Compliance with Movement Behavior Guidelines and Obesity among Malaysian Preschoolers

**DOI:** 10.3390/ijerph18094611

**Published:** 2021-04-27

**Authors:** Shoo Thien Lee, Jyh Eiin Wong, Geraldine K. L. Chan, Bee Koon Poh

**Affiliations:** 1Center for Community Health Studies (ReaCH), Faculty of Health Sciences, Universiti Kebangsaan Malaysia, Kuala Lumpur 50300, Malaysia; shoothien@hotmail.com (S.T.L.); wjeiin@ukm.edu.my (J.E.W.); 2Center for Research in Development, Social and Environment (SEEDS), Faculty of Social Sciences and Humanities, Universiti Kebangsaan Malaysia, Bangi 43600, Selangor, Malaysia; geralckl@ukm.edu.my

**Keywords:** movement behavior, obesity, physical activity, preschoolers, screen time, sleep

## Abstract

A daily balance of physical activities, sedentary behaviors and sleep are important for maintaining the health of young children. The aim of this study is to explore the association between 24-h movement behavior of Malaysian children aged 4 to 6 years with weight status. A total of 230 preschoolers were recruited from 22 kindergartens in Kuala Lumpur. Physical activity was assessed by Actical accelerometer while screen time and sleep duration were proxy-reported by parents. Children spent on average 5.5 ± 1.3 h on total physical activity (including 1.0 ± 0.4 h of moderate- vigorous physical activity), 3.0 ± 1.6 h on screen activities and 9.5 ± 1.3 h sleeping daily. The proportion of children who complied with physical activity and sleep guidelines were 48.7% and 55.2%, respectively. About 25.2% of children met screen time recommendation. Only 6.5% of children met all three age-specific physical activity, screen time and sleep guidelines. Children who met any two guidelines were less likely to be overweight or obesity compared to those who did not meet any of the guidelines (OR: 0.276; 95% CI: 0.080–0.950). In conclusion, Malaysian preschoolers have low compliance to movement behavior guidelines, especially in meeting screen time recommendations. Compliance to movement behavior guidelines was associated with lower odds of overweight and obesity.

## 1. Introduction

Childhood obesity potentially leads to a number of health problems such as cardiovascular diseases, metabolic syndrome, psychological disorders, impaired bone health, diabetes and hypertension [1,2]. Children with obesity have a higher risk of overweight or obesity later during their adolescent years [3] and adulthood [4]. In 2016, the World Health Organization reported that more than 340 million children aged 5 to 19 years were overweight and obese [5]. In 2018, an estimated 40 million children under five years old were reportedly overweight with nearly half of these children living in Asian countries [5]. Among the East and Southeast Asian countries, Malaysia has the highest prevalence of childhood obesity [6], with 16% of preschoolers aged 4 to 6 years reported to be overweight or obesity [7]. The National Health and Morbidity Survey (NHMS) had reported that the prevalence of childhood obesity in Malaysia has been increasing over the years, with 6.1% in 2011, 11.9% in 2015 and 14.8% in 2019 [8].

Childhood obesity could be reduced by multi-component modifiable behavioral changes, including physical activity, sedentary behavior and dietary patterns [9]. Several studies have reported that a combination of physical activity, sedentary time and sleep duration is important in maintaining the health status of children [10,11,12]. High physical activity, less sedentary behavior and sufficient sleep time is beneficial in maintaining a healthy body weight and reducing child adiposity [13,14]. Importantly, two systematic reviews have concluded that healthy movement behaviors in early childhood track through middle childhood to adolescence [15,16]. Additionally, these healthy behaviors also improve motor development, cardiometabolic health and cardiorespiratory fitness in children and adolescents [13,14,17]. For instance, Japanese children aged 3 to 5 years who do not meet all these healthy behavior guidelines had higher odds (OR: 1.139; 95% CI: 1.009–1.285) of overweight and obesity [18] compared to those who meet the guidelines. Another study among Chinese children aged 4 to 6 years showed that screen time is associated with high odds (OR: 3.76; 95% CI: 1.50–9.45) of overweight and obesity [19]. Several studies among young children aged 1 to 5 years, however, reported that compliance to these healthy behavior guidelines is not associated with body weight status and adiposity [20,21,22]. These contradictory findings may be due to the nature of adiposity at this young age, whereby it is too early to be certain of excess adiposity in young preschoolers [21]. Thus, further studies are needed to understand the links between healthy behaviors and adiposity rebound in childhood.

In Malaysia, school-aged children and adolescents reportedly engage in low levels of physical activity and high levels of screen time [23]. Moreover, physical activity and screen activities of Malaysian school-aged children have been associated with obesity [24]. The beneficial effects of physical activity, sedentary behavior and sleep on preschoolers’ body weight status are mostly reported by studies conducted in high income countries such as Sweden, Greece, USA, Japan and Canada [13,18,20,21]. There is a lack of similar studies conducted in low- and middle-income countries, especially among the preschooler age group. Malaysia, being an upper-middle income country [25] that comprises multiethnic and multicultural communities, is best suited to fill this gap. Thus, the aims of this study are to determine (i) the physical activity, screen activities and sleep time of Malaysian preschoolers of various ethnicities, and (ii) the association of these behaviors with their body weight status. Compliance with movement behavior, i.e., meeting physical activity, screen time and/or sleep time guidelines, is hypothesized to be associated with healthier weight statuses.

## 2. Methodology

### 2.1. Study Design and Participants

This cross-sectional study was conducted from January until November 2017. Ethical approval was obtained from the Research Ethics Committee of Universiti Kebangsaan Malaysia (Code: NN-2017–174). This study also obtained the permission of the Ministry of Education (MOE), the Community Development Department (KEMAS) under the Ministry of Rural Development, and the respective principals of the preschools involved. Written, informed consent from parents and verbal assent from preschoolers were obtained before conducting the study.

The subjects of this study were recruited from a total of 17 government and 5 private preschools in Kuala Lumpur. In Malaysia, formal preschool education caters to children aged 4 to 6 years and is operated by government departments, such as MOE, KEMAS and the Ministry of National Unity (*Perpaduan*), as well as private entities [26]. Government preschools operate within primary school compounds (under MOE) or community buildings and halls (under KEMAS and *Perpaduan*), while private preschools operate at shop lots, complex or dwelling-based buildings (bungalow or corner house units) [27,28]. Government and private preschools with dwelling-based design usually provide outdoor playground and sport facilities for children’s playtime [27,28]. The majority of government preschools are small buildings with shared facilities and have been rated fair for their physical environment using Children Physical Environment Rating Scale 5 [27]. Private preschools located at dwelling-based buildings usually provide better space and physical environment than preschools located at shop lots [28].

A total of 230 preschoolers joined this study. The inclusion criteria were preschoolers who were: (i) aged 4 to 6 years old; (ii) from Malay, Chinese or Indian ethnic groups (the three main ethnicities of Malaysia); and (iii) attending preschool. Preschoolers who had physical or mental disability, or who were either sick or absent during data collection, were excluded from this study.

### 2.2. Physical Activity Assessment

Physical activity was assessed using Actical accelerometers (Mini Mitter Respironics Inc., Bend, OR, USA). The epoch time used for this study was 15 s. Each Actical was worn on an elastic belt, positioned around the waist over the right hip [21]. Written instructions with pictures illustrating Actical wearing procedures were given to parents. Parents were also given face-to-face briefings on the Actical wearing, and reminded daily through private messaging to wear the Actical on their child. Preschoolers wore the Actical for seven consecutive days, except during shower, swimming, water-play activities and sleep. Each subject’s Actical valid wear time must fulfill at least three weekdays and one weekend day. A minimum of 10 h of wear time was required for each day to be considered valid [29]. The valid wear time in this study was defined as the total Actical wear time minus non-wear time. Non-wear time is indicated by at least 60 min of consecutive zero activity count, with the allowance of one to two minutes of activity counts (0–100) [21,29].

Adolph15 cut-off was used to classify physical activity into three levels: Moderate to vigorous (MVPA) (activity counts ≥287), light (6< activity counts <287) and sedentary (activity counts ≤6) [30,31]. Adolph15 cut-off was employed in this study as it had been validated by calorimetry and observation studies [31]. The total physical activity level is the sum of average daily MVPA and light-intensity activity. Three guidelines were referred to in this study: World Health Organization (WHO) 2019 Guidelines on Physical Activity, Sedentary Behaviour and Sleep for Children under 5 years [32], WHO 2020 Guidelines on Physical Activity and Sedentary Behaviour for Children and Adolescents (5–17 years) [33] and Canadian 24-h Movement Guidelines for Children and Youth for children aged 5 to 6 years old [34]. Children aged 4 years were considered to have met the physical activity guidelines if they recorded a total physical activity of ≥3 h; and with at least an hour of MVPA [32]. Children aged 5 to 6 years were classified as meeting physical activity guidelines if they spent at least 60 min per day on MVPA [33].

### 2.3. Screen Time, Sleep Assessment and Sociodemography

Children’s screen activities include watching television/videos/DVD, using computer/internet/smartphone/tablet and playing video games. Parents reported children’s time (in minutes) spent on each screen activities on both weekdays and weekend day separately. The time spent on screen activities was calculated as a weighted average over weekdays (total of minutes during weekdays × 5) and weekend days (total of minutes during weekend × 2). The recommended daily screen time is ≤60 min for children aged 4 years [32]; and ≤120 min for children aged 5 to 6 years [34].

Sleep time including daytime naps and nighttime sleep were proxy-reported by parents in a questionnaire, which was adapted from the Healthy Active Preschool and Primary Years (HAPPY) Study [35]. The sleep time questions were face validated via cognitive interviews with 14 mothers [36]. The recommended daily sleep time is 10 to 13 h for children aged 4 years [32]; and 9 to 11 h for children aged 5 to 6 years [34]. Based on the recommended guidelines, children were classified into two groups, either meeting sleep time or not meeting sleep time recommendations.

Socio-demographic characteristics including age, sex, ethnicities, parental education and household income were reported by parents through a self-administered pen-and-paper questionnaire. Monthly household income in Malaysian Ringgit (MYR) was categorized into three groups using the following categories: low (<4850 MYR); middle (between 4850 MYR and 10,959 MYR); and high (≥10,960 MYR) [37].

### 2.4. Anthropometric Measurements

All anthropometric measurements were conducted by a researcher, who is a certified anthropometrist (Level 1) by the International Society for the Advancement of Kinanthropometry (ISAK). Standing height was measured with a wall-mounted stadiometer (SECA model 213, Hamburg, Germany) to the nearest 0.1 cm. Weight was measured to the nearest 0.1 kg with a digital weighing scale (SECA model 803, Hamburg, Germany). Body mass index (BMI, kg/m^2^) was calculated by dividing weight (kg) by the square of height (m). BMI-for age z-score (BAZ), height-for-age z-score (HAZ) and weight-for-age z-score (WAZ) were calculated using the WHO AnthroPlus software version 1.0.3 [38]. Based on the WHO Child Growth Standards 2006 [39] and WHO Growth Reference 2007 [40], preschoolers were categorized into thinness, normal weight, overweight or obese groups using BAZ values. The cut off for stunting was HAZ < − 2SD [39,40].

### 2.5. Statistical Analysis

The statistical analysis was conducted by using SPSS version 22.0 (IBM Corporation, Chicago, IL, USA). Descriptive analysis was used to analyze the children’s characteristics, body weight status, physical activity and time spent on sedentary activities and screen time. Sex differences in anthropometric measurements, physical activity, screen time and sleep time were examined using t-tests. Differences in socio-demographic characteristics and body weight status between boys and girls were tested by Chi-square tests. One-way ANOVA and Tukey post-hoc tests were performed to determine the differences in time spent on physical activity, screen activities and sleep by age groups, socio-demographic groups and body weight status. Five logistic regression models with the “Enter” fitting option was conducted to determine the odds ratio of being overweight or obese. Body weight status groups were combined into two groups, defined as overweight/obese and normal weight/thinness (reference group) in the logistic regression models. The independent variables of logistic model 1 was based on four groups of meeting movement behavior guidelines (i.e., did not meet any guideline, met 1 guideline, met 2 guidelines and met 3 guidelines); while logistic model 2 used all individual and combinations of guidelines’ compliance. The independent variables used in logistic models 3 to 5 were binary (model 3: met physical activity guideline, model 4: met screen time guidelines and model 5: met sleep time guidelines). All logistic models were 85.2% accurate in body weight prediction and showed good fit for the data as confirmed by the Hosmer and Lemeshow tests (*p* > 0.05). Confounding variables, including sex, ethnicity and household income, were adjusted in all logistic regression models. Values were reported as the mean ± standard deviation or number of subjects (percentage). Significance level is set at *p* < 0.05.

## 3. Results

At the start, a total of 283 children consented to take part in the study, but only 205 children (72%) completed Actical wearing time with at least three weekdays and one weekend day. Among the children who did not have complete Actical data, a total of 56 children repeated the Actical wearing, with only 35 who completed the Actical wearing the second time around. In total, there were 240 children who fulfilled the Actical valid wear time; however, 10 children were excluded from this analysis due to incomplete questionnaire data. Thus, the final number of subjects included in data analysis was 230.

Table 1 shows the socio-demographic characteristics and anthropometry of children. Majority of the children were Malays (40.4%), had parents who had completed secondary education (maternal: 58.6% and paternal: 60.4%), and were from low household income families (68.4%). The socio-demographic characteristics were not significantly different between boys and girls. Mean age of the children who participated in this study was 5.51 ± 0.64 years. Mean score of BAZ, HAZ and WAZ of children were −0.14 ± 1.53, −0.48 ± 1.17 and −0.38 ± 1.53, respectively. There were no differences in the anthropometric measurements of boys and girls. Prevalence of overweight and obesity was 15.3%, with 18.0% in boys and 12.6% in girls, and the percentage difference was not statistically significant (*p* = 0.253). About 4.7% and 7.4% of children had thinness and stunting, respectively.

Table 2 shows that the children spent on average 5.5 h on physical activity, including time spent on MVPA. They spent up to three hours of daily time on screen activities, mostly watching television (2.2 h). Malay children had an hour longer screen time than Chinese children, and a half hour longer screen time than Indian children (*p* < 0.05). Mean sleep time of children was 9.5 h, including 1.2 h of day-time nap. Children with obesity had on average an hour less sleep time as compared to children who are normal weight or overweight (*p* < 0.05). Although not statistically significant, children with obesity tended to have low physical activity level (*p* = 0.192) and high screen time (*p* = 0.304) compared to children with other weight statuses. However, this pattern was not observed in children who are overweight. Children who are overweight spent similar amounts of time on physical activity, screen activities and sleep as children with normal weight.

The Venn diagram shows that nearly half (48.7%) of the children met the physical activity guidelines (Figure 1). More than half of the children (55.2%) met the sleep guidelines while a quarter (25.2%) of the children met the screen time guidelines. A total of 17% of preschoolers did not meet any of these behavior guidelines; while only 6.5% of preschoolers met all three guidelines for physical activity, screen time and sleep time.

Table 3 suggests that the odds of children having overweight or obesity is based on adherence to movement behavior guidelines. Children who met two guidelines had lower odds of overweight or obesity (OR = 0.276; 95%CI= 0.080–0.950) (Model 1) than those who did not meet any of the guidelines. Specifically, meeting the combination of physical activity and sleep guidelines was associated with lower odds of overweight or obesity (OR = 0.178; 95%CI = 0.033–0.946) (Model 2) than not meeting any of the guidelines.

## 4. Discussion

The present study reported that Malaysian preschoolers who met movement behavior guidelines had lower odds of overweight or obesity compared to those who did not meet any of the guidelines. This suggests that compliance to movement behavior guidelines was associated with lower odds of overweight or obesity. This finding is consistent with another study which reported that Japanese preschool children who failed to meet movement behavior guidelines were more likely to experience overweight or obesity as compared to children who met all guidelines (OR: 1.139; 95% CI: 1.009–1.285) [18]. Preschoolers in China also showed lower odds for overweight and obesity when meeting screen time guidelines compared to those who did not meet screen time guidelines [19]. In contrast, studies conducted among preschoolers and toddlers in Sweden, Canada and Australia found that compliance with movement behavior guidelines was not associated with body weight status [20,21,41]. This was also supported by a prospective longitudinal study, which reported that compliance with healthy movement behavior guidelines among toddlers and preschoolers was not related to BAZ and body composition at the age of 5 years [22].

The association between meeting 24-h movement guidelines and body weight status seem to corroborate with findings of those among school-aged children and adolescents [11,42]. A study by Roman–Viñas et al., which involved 6128 children (9 to 12 years old) recruited from 12 countries, reported that the odds for obesity was the lowest when children met all three movement behavior recommendations [11]. A Canadian study of 4157 children and adolescents (6 to 17 years old) found that meeting none, one or two recommendations were associated with higher BAZ and waist circumference, as compared to meeting all three recommendations [42]. In addition, meeting fewer recommendations were also associated with higher systolic blood pressure, insulin, triglycerides and lower high-density-lipoprotein (HDL) cholesterol levels [42].

The present study found that meeting two movement behavior recommendations instead of meeting all recommendations was associated with lower odds of overweight or obesity. The low number of children who met all three recommendations (*n* = 15) rendered insufficient power to detect significant association with childhood obesity [43]. Of the children who met all three recommendations, there were only five (33.3%) who were overweight or obese. In addition, Fisher Exact test found no association between compliance of three recommendations with obesity (*p* = 0.059). Hence, we were not able to determine the association between children meeting all three guidelines with childhood obesity status. In Model 2, we found that only the combination of physical activity and sleep recommendations was associated with lower odds of overweight or obesity. However, it is notable that half of the subgroups in Model 2 had low numbers of children (*n* < 20). As such, most subgroups were inadequately powered to demonstrate the association with body weight status.

The present study found that the prevalence of obesity among preschoolers was 15.3%. This prevalence was similar to those reported by the South East Asian Nutrition Survey (SEANUTS) Malaysia (16% of urban children aged 4 to 6 years) [7] and NHMS 2019 (14.8% of children aged 5 to 17 years) [8]. The prevalence of thinness and stunting in this study was 4.7% and 7.4%, respectively. SEANUTS Malaysia reported similar prevalence of thinness and stunting among urban children aged 4 to 6 years, which was 5.0% and 7.3%, respectively [7]. The NHMS 2019, on the other hand, assessed thinness and stunting among children aged 5 to 17 years, and reported that prevalence was, respectively, 9.6% and 11.3% from urban; and 11.2% and 17.1% from rural areas of Malaysia [8]. The lower prevalence of thinness and stunting found in the present study as compared to NHMS 2019 may be due to lower thinness and stunting problems among children living in the urban capital city of Malaysia [8].

This study provides the first description on Malaysian preschoolers’ compliance with the 24-h movement guidelines. The key finding indicates that up to 17% of preschoolers did not meet any of the physical activity, sedentary screen time and sleep time guidelines. This percentage was the highest as compared to studies conducted in Singapore (11.2%) [44], Australia (7.4%) [45], Japan (3.6%) [18], Canada (3%) [21], Belgium (2.9%) [46], China (2.7%) [19] and Sweden (0.5%) [20]. The present study found that only a very small percentage (6.5%) of preschoolers in Kuala Lumpur met all three guidelines. This compliance rate was similar to studies conducted in Singapore (5.5%) and Belgium (5.6%) but was lower when compared with those reported in Japan (21.5%), Sweden (18.4%), China (15.0%), Australia (14.9%), Canada (12.7%) and New Zealand (boys: 7.0%; girls: 9.3%) [22]. Socio-economic background could be a factor that influences the movement behavior of children [47]. Malaysia is an upper-middle income country, whereas all of the other studies were conducted in high income countries [25], except for Guan and colleagues’ study, which was conducted in Beijing, a high-income city [48].

The present study shows that one in two children met the physical activity guidelines. This compliance is in the middle range as compared to studies conducted in Australia (93.1%) [45], Japan (91.6%) [18], USA (91.5%) [47], New Zealand (boys: 83.6%; girls: 79.4%) [22], China (65.4%) [19], Canada (61.8%) [21], Singapore (59.6%) [44], Sweden (31.0%) [20] and Belgium (11.0%) [46]. The variation in physical activity guidelines compliance may be explained by the use of (i) different measurement instruments, (ii) different cut-offs to classify activity levels, (iii) different epoch length for accelerometer and (iv) different accelerometer wear position. Physical activity compliance in the present study was comparable to the Canadian results [21], where both studies used Actical accelerometers. The very low physical activity compliance in the study from Belgium could be due to the use of a higher cut point classification [46]. More light physical activity was captured instead of sedentary behavior when using a longer epoch length [49]. A Swedish study used a shorter epoch length as compared to other studies (10 s versus 15 s) [20], thus leading to less total physical activity in comparison. The accelerometers were worn on the waist of children in all of the above-mentioned studies with the exception of the study conducted among Singaporean children in which accelerometer was worn on the wrist [44].

The reported screen time of preschoolers in this study was double that reported in the nationwide SEANUTS Malaysia study (3 h versus 1.5 h) conducted a decade ago [50]. A possible reason is the increasing availability and accessibility of screen media, such as television, smartphone, tablet, computer or other digital devices [51,52]. Despite the availability of various media devices, watching television was still the main screen activities of preschool children; it contributed to 75% of screen time in the present study. Three out of four preschoolers (74.8%) had exceeded the recommended screen time. The low compliance of screen time guidelines might possibly be due to the absence of rules imposed on screen activities at home. Parental self-efficacy to limit screen time and in practicing screen time rules will aid in reducing screen time of children [35,53]. Moreover, screen time of young children (0 to 8 years old) was strongly associated with parental screen activities and parental attitude towards screen activities [54,55]. Parents and caregivers are inclined to engage their children with screen activities to create uninterrupted time for themselves, for example, when they are busy with household chores [51,56].

The present study shows that only about half of the Malaysian preschoolers met the sleep time recommendation. This compliance rate was lower compared to studies conducted in Australia [45], Canada [21], Sweden [20], Japan [18], Belgium [46], USA [47] and New Zealand [22], which showed that at least 80% of their preschoolers met daily sleep time recommendation. However, sleep time compliance in China (29.5%) [19] and Singapore (13.7%) [44] were worse than Malaysian children (55.2%). In terms of sleep duration, the present study found that Malaysian preschoolers had lower sleep duration (9.5 h) as compared to the above-mentioned studies, which reported sleep time ranging from 10.5 to 11.0 h, except for China (9.6 h) and Singapore (9 h). The variance in sleep time between Asian and Western countries may be due to different cultural practices in relation to night-time sleep and day-time napping behaviors among preschoolers [44].

In year 2011–2012, a comparison study on preschoolers’ sleep behavior was conducted among 13 countries, including Malaysia [57]. As compared to Mindell and colleagues study [57], the present findings show that Malaysian preschoolers had lower total sleep duration (9.5 h versus 10.8 h), daily nap time (1.2 h versus 1.7 h) and night-time sleep (8.3 h versus 9.1 h). The shorter sleep time reported in this present study could be due to the slightly older age group as compared with Mindell’s study (4–6 years versus 3–6 years) or due to changing sleep patterns in relation to physical activity transition with time [58]. In the present study, one possible explanation of shorter sleep hours was children spending more time in other activities. This is because physical activity, sedentary behavior and sleep are co-dependent and form part of a 24-h period [59]. A cohort study found that longer screen activities in early life was associated with shorter sleep time and physical activity in preschoolers [60]. Moreover, the shorter sleep duration of preschoolers was possibly due to inconsistencies of bedtime routines. A previous study showed that only 62.0% of Malaysian preschoolers had consistent bedtime routines [57]. This percentage was lower as compared to preschoolers in Australia (89.5%), Canada (85.7%), United Kingdom (91.9%), USA (81.6%) and Singapore (79.0%) [57]. A consistent bedtime routine is important to improve sleep latency and reduce sleep problems for children [61]. Hence, with appropriate parenting styles, having consistent bedtime routine and early bedtime is important to be practiced from a young age [57,62].

This study highlights that compliance to movement behavior guidelines was associated with a healthier body weight status among young children. Obesity intervention programs for young children should establish healthy movement behaviors, such as high physical activity level, low screen time and adequate sleep duration. Strategies to increase physical activity should consider parental support on active play and role modeling [63]. Strategies to reduce screen time may include substitution of screen activities with creative active games and practicing screen time rules from a young age [55]. Strategies to promote consistent bedtime routines and early bedtimes, are important to increase sleep duration and sleep quality [57,62]. Future prospective studies are needed to capture the behavioral and body weight changes from young childhood to adolescence. Studies on determinants of movement behaviors that encompass intrapersonal, interpersonal, organizational, community and policy levels [63], are important to inform the development of an effective intervention program.

This is the first study conducted in Malaysia that descriptively reported preschoolers’ compliance of 24-h movement behavior guidelines and its association with body weight status. In addition, the subjects who participated in this study represented the three main ethnicities and were from low, middle to high socio-economic backgrounds. The main limitation of this study was that screen time and sleep time were measured subjectively through proxy-reporters, where possible errors might be influenced by social desirability when reporting as well as the recall bias of parents. Further studies may consider objectively measuring whole day (24 h) activities including sleep time by using accelerometer. Furthermore, the association analyses should also be replicated in a large cohort of children to verify the observations made in this initial cohort. The findings of this study are not generalizable to the entire Malaysian preschoolers’ population as the subjects were recruited only from the capital city of Kuala Lumpur, where physical activity and sedentary behaviors could be different from those of children living in other parts of the country.

## 5. Conclusions

Low compliance to movement behavior guidelines was found among Malaysian preschoolers, particularly in achieving screen time recommendations. Meeting physical activity, screen time and sleep duration guidelines were associated with lower odds of overweight or obesity. As such, suitable strategies and actions are needed to promote the movement behavior guidelines, in particular guidance for less screen time, to parents and the community at large. This will ensure an increasing awareness of the importance of physical activity and sleep, and reducing sedentary behavior, as well as aid in promoting Malaysian preschoolers’ compliance to the guidelines, particularly in limiting screen activities. Future studies looking at the association of movement behaviors with other obesity indicators, including metabolic syndrome and adiposity, should be considered, especially among young childhood populations in Asia, where data is lacking.

## Figures and Tables

**Figure 1 ijerph-18-04611-f001:**
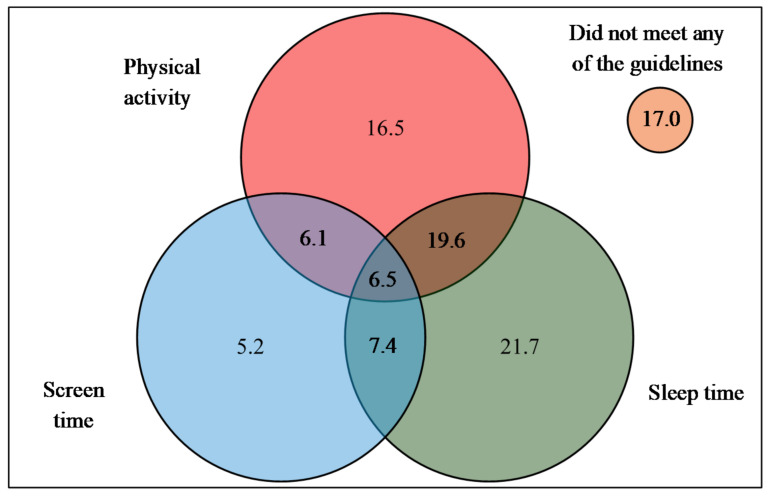
Proportion of preschoolers meeting physical activity, screen time and sleep guidelines. Met physical activity guidelines: ≥3 h total physical activity with at least an hour of MVPA for children aged 4 years and ≥an hour of MVPA for children aged 5 to 6 years; met screen time guidelines: ≤60 min for children aged 4 years and ≤120 min for children aged 5 to 6 years; and met sleep time guidelines: 10–13 h for children aged 4 years and 9–11 h for children aged 5 to 6 years.

**Table 1 ijerph-18-04611-t001:** Anthropometric and socio-demographic characteristics of preschoolers (*n* = 230).

Characteristics	Boys (*n* = 119)	Girls (*n* = 111)	All (*n* = 230)
	*n* (%)	*n* (%)	*n* (%)
Ethnicities			
Malay	46 (41.5)	47 (39.5)	93 (40.4)
Chinese	33 (29.7)	37 (31.1)	70 (30.4)
Indian	32 (28.8)	35 (29.4)	67 (29.2)
Maternal education level			
Primary	4 (3.7)	5 (4.2)	9 (4.0)
Secondary	59 (54.1)	74 (62.7)	133 (58.6)
Tertiary	46 (42.2)	39 (33.1)	85 (37.4)
Paternal education level			
Primary	8 (7.5)	5 (4.5)	13 (6.0)
Secondary	60 (56.1)	71 (64.5)	131 (60.4)
Tertiary	39 (36.4)	34 (31.0)	73 (33.6)
Household income in MYR ^#^			
Low (<4850)	72 (64.9)	85 (71.4)	157 (68.3)
Middle (4850–10,959)	34 (30.6)	27 (22.7)	61 (26.5)
High (≥10,960)	5 (4.5)	7 (5.9)	12 (5.2)
Body weight status			
Thinness	6 (5.4)	5 (4.2)	11 (4.7)
Normal weight	85 (76.6)	99 (83.2)	184 (80.0)
Overweight	9 (8.1)	7 (5.9)	16 (7.0)
Obese	11 (9.9)	8 (6.7)	19 (8.3)
Stunting			
Stunted	8 (7.2)	9 (7.6)	17 (7.4)
Not stunted	111 (92.8)	102 (92.4)	213 (92.6)
	Mean ± SD	Mean ± SD	Mean ± SD
Age	5.58 ± 0.59	5.44 ± 0.68	5.51 ± 0.64
Weight (kg)	19.5 ± 5.5	18.3 ± 4.5	18.9 ± 5.1
Height (cm)	111.1 ± 6.5	109.3 ± 7.2	110.2 ± 6.9
BMI (kg/m^2^)	15.60 ± 2.87	15.15 ± 2.27	15.37 ± 2.59
BAZ (z-score)	−0.01 ± 1.74	−0.26 ± 1.30	−0.14 ± 1.53
HAZ (z-score)	−0.47 ± 1.19	−0.50 ± 1.16	−0.48 ± 1.17
WAZ (z-score)	−0.28 ± 1.69	−0.48 ± 1.38	−0.38 ± 1.53

Abbreviation: SD, standard deviation; MYR, Malaysian ringgit; BMI, body mass index; BAZ, body mass index-for-age z score; HAZ, height-for-age z score; WAZ, weight-for-age z score; No difference in proportion distribution of children within socio-demographic groups and body weight status by using chi-square test; No difference in anthropometric measurements between boys and girls by using independent-*t*-test; and ^#^ 1 US dollar = 4.04 MYR (as of 30 January 2021).

**Table 2 ijerph-18-04611-t002:** Mean time spent on physical activity, screen activities and sleep (mean ± SD).

Characteristics	Physical Activity (Minutes)	Watching Television (Minutes)	Playing Computer/ Smartphone/ Tablets/ Video Games (Minutes)	Total Screen Time (Minutes)	Overnight Sleep (Hours)	Daily Nap (Hours)	Total Sleep Time (Hours)
All	330 ± 77	133 ± 75	44 ± 53	177 ± 94	8.3 ± 1.1	1.2 ± 0.9	9.5 ± 1.3
Age groups							
4 + years	331 ± 75	133 ± 76	41 ± 51	174 ± 95	8.3 ± 1.1	1.3 ± 0.9	9.6 ± 1.4
5 + years	329 ± 80	139 ± 80	49 ± 58	188 ± 102	8.2 ± 1.1	1.1 ± 0.9	9.3 ± 1.2
6 + years	332 ± 75	120 ± 63	35 ± 44	156 ± 74	8.5 ± 1.1	1.0 ± 0.8	9.6 ± 1.3
Sex							
Boys	340 ± 78	132 ± 76	43 ± 50	175 ± 98	8.4 ± 1.0	1.0 ± 0.9 ^a^	9.4 ± 1.2
Girls	321 ± 76	133 ± 75	45 ± 56	178 ± 91	8.2 ± 1.2	1.3 ± 0.8 ^b^	9.5 ± 1.4
Ethnicities							
Malay	335 ± 80	156 ± 83 ^a^	50 ± 59	206 ± 100 ^a^	8.4 ± 1.1 ^a^	1.3 ± 0.9	9.6 ± 1.4
Chinese	329 ± 77	98 ± 65 ^b^	45 ± 54	143 ± 92 ^b^	8.6 ± 1.0 ^a^	1.0 ± 0.8	9.5 ± 1.2
Indian	325 ± 74	137 ± 60 a	33 ± 43	170 ± 76 ^b^	7.9 ± 1.0 ^b^	1.2 ± 0.8	9.2 ± 1.2
Maternal education level							
Primary	342 ± 90	113 ± 67	45 ± 61	158 ± 111	7.8 ± 1.7	1.4 ± 0.9	9.2 ± 1.8
Secondary	333 ± 74	140 ± 79	40 ± 50	180 ± 93	8.3 ± 1.0	1.3 ± 0.9	9.6 ± 1.3
Tertiary	323 ± 79	124 ± 69	50 ± 58	174 ± 96	8.4 ± 1.2	0.9 ± 0.7	9.2 ± 1.1
Paternal education level							
Primary	333 ± 91	114 ± 63	52 ± 47	166 ± 82	8.7 ± 1.0	0.7 ± 0.5	9.4 ± 1.1
Secondary	332 ± 73	143 ± 75	37 ± 44	180 ± 91	8.3 ± 1.0	1.3 ± 0.9	9.5 ± 1.3
Tertiary	327 ± 80	116 ± 75	52 ± 65	168 ± 103	8.4 ± 1.2	1.0 ± 0.8	9.3 ± 1.3
Household income in MYR ^#^							
Low (<4850)	331 ± 78	142 ± 74 ^a^	40 ± 52	184 ± 92	8.3 ± 1.1	1.3 ± 0.9 ^a^	9.6 ± 1.3
Middle (4850–10,959)	327 ± 82	121 ± 72 ^a^	50 ± 54	168 ± 91	8.3 ± 1.1	0.9 ± 0.8 ^b^	9.2 ± 1.2
High (≥10,960)	328 ± 49	73 ± 69 ^b^	53 ± 69	126 ± 130	8.5 ± 1.2	0.7 ± 0.6 ^b^	9.2 ± 1.4
Body weight status							
Thinness	349 ± 85	155 ± 68	37 ± 56	192 ± 102	7.9 ± 0.9	1.2 ± 0.5	9.1 ± 1.2 ^ab^
Normal weight	330 ± 75	130 ± 76	42 ± 51	172 ± 91	8.4 ± 1.1	1.2 ± 0.9	9.6 ± 1.3 ^a^
Overweight	351 ± 85	117 ± 72	55 ± 53	172 ± 101	8.5 ± 1.2	1.2 ± 1.0	9.7 ± 1.2 ^a^
Obese	300 ± 89	156 ± 68	57 ± 73	213 ± 111	7.8 ± 0.9	0.9 ± 0.7	8.7 ± 1.2 ^b^

Abbreviation: SD, standard deviation; MYR, Malaysian ringgit; ^ab^ Different superscript indicates significant difference (*p* < 0.05) within the socio-demographic groups and body weight status by using One-way ANOVA test and Tukey post hoc test; and ^#^ 1 US dollar = 4.04 MYR (as of 30 January 2021).

**Table 3 ijerph-18-04611-t003:** Association of movement guidelines compliance with overweight or obese status.

Logistic Regression Models	Thinness/Normal Weight (*n*)	Overweight/ Obese (*n*)	OR	95% CI	*p*-Value
Model 1 ^#^					
Met all recommendations	10	5	2.268	0.488–10.547	0.296
Met any two recommendations	70	6	0.276 *	0.080–0.950	0.041
Met one recommendation	84	16	0.593	0.212–1.659	0.319
Not meeting any recommendations	31	8	1.000		
Model 2 ^#^					
Physical activity + screen time + sleep time	10	5	2.225	0.475–10.426	0.310
Physical activity + screen time	13	1	0.223	0.024–2.046	0.185
Physical activity + sleep time	43	2	0.178 *	0.033–0.946	0.043
Screen time + sleep time	14	3	0.757	0.131–4.365	0.755
Physical activity	32	6	0.531	0.145–1.943	0.339
Screen time	11	1	0.308	0.039–2.872	0.310
Sleep time	41	9	0.772	0.250– 2.382	0.652
Not meeting any recommendations	31	8	1.000		
Model 3 ^#^					
Met physical activity recommendation	98	14	0.606	0.273–1.344	0.218
Not meet physical activity	97	21	1.000		
Model 4 ^#^					
Met screen time recommendation	48	10	0.952	0.368–2.463	0.919
Not meet screen time	147	25	1.000		
Model 5 ^#^					
Met sleep time recommendation	108	19	1.141	0.515–2.528	0.745
Not meet sleep time	87	16	1.000		

Abbreviation: OR, odds ratio of overweight or obese; CI, confidence interval; ^#^ All models were adjusted for sex, ethnicity and household income; and * Significance at *p* < 0.05.

## Data Availability

The datasets analyzed during the current study are not publicly available due to confidentiality privacy of the subjects, but can be made available by the corresponding author upon reasonable request.

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
