# Peer review of "Association between Compliance with Movement Behavior Guidelines and Obesity among Malaysian Preschoolers"

_ijerph, 2021, doi:10.3390/ijerph18094611_

Round 1

Reviewer 1 Report

Thank you for the opportunity to re-review this paper. The authors did a nice job updating the manuscript but I still have concerns and comments on the paper.

Abstract: Reverse the direction here, it should read how many met the guideline to match the other sentences. About 74.8% of children exceeded screen time recommendation

Lines 54-55: “us, more similar studies are needed to verify the association of healthy behaviours with body weight status and adiposity among young children” A key component that is missing is understanding the adiposity rebound that occurs within this age range. That is a key factor that is not mentioned or considered within this assessment of cross-sectional literature. The current study has a slightly older sample (4-6y) and may not be subject this these phenomena. This consideration should be addressed, rather than focusing on verifying results.

Line 65-66: “Compliance with movement behaviour is hypothesized to be associated with healthy weight status.” Please clarify this hypothesis, as in compliance with which behaviors, or how many guidelines met. The authors mention multiple associations within the introduction.

Line 75-77:  Please expand upon whether kindergarten is formal schooling, such as beginning grade school, or if this structure or like childcare settings in other countries. The type of settings has implications for their physical activity included within the day. This is confusing because later the manuscript requires them be from a kindergarten (line 82), but before it is described as formal preschool education (line 75). If I am understanding correctly, it seems the authors are using preschool and kindergarten interchangeably. If so, please rephrase line 75 “Formal preschool education (herein: kindergarten) in Malaysia caters for children aged 4 to 6 years and is operated by government departments, such as the Ministry of Education (MOE),”.

Then add a sentence to clarify what the classroom is equivalent to in other countries (kindergarten or preschool settings).

Lines 79-82 confuse enrollment vs inclusion of ethnicity. Suggest rephase to clarify this distinction:

“A total of 230 preschoolers joined this study. The inclusion criteria were preschoolers who were: (i) 4 to 6 years old; (ii) from Malay, Chinese or Indian ethnic groups (the three main ethnicities of Malaysia); and (iii) attending kindergarten.”

Lines 99-105 should be included in the results.

Lines 112-113, suggest new WHO Guidelines 2020 WHO guidelines on physical activity and sedentary behaviour for children and adolescents aged 5–17 years: summary of the evidence | International Journal of Behavioral Nutrition and Physical Activity | Full Text (biomedcentral.com)

Line 125-126: Please clarify the amount the parent reported, did they report per weekday and weekend for each device? Or a weekly average?

Line 178. The standard deviation of age is not included, and only shows the tenths, whereas the table is to the hundredths.

Line 180: Suggest delete “There were no differences in anthropometric measures of boys and girls”, as the next sentence shows there is a difference.

Differences between age that were explored should be reported in statistical analysis (Suggest with sex differences, line 155 or clarify this in line 159) and mentioned in results.

Lines 192-195, add p-value for comparisons

Line 196-197 Delete as the results are not significant, or show p-value for comparisons.

Line 199-202 It is unclear if these are significant comparisons. Please add p-value.

Table 3 is unclear and confusing. Suggest update Table 3 so the column reads from top to bottom met all guidelines, met two guidelines, met one guideline, met 0 guidelines. A similar schematic should be shown for Model 2 that goes from meeting all three guidelines to two guidelines, then one guideline, and zero guidelines. Suggest include proportion thinn/normal and overweight/obese in a supplementary table or other means, as reads as though the authors are running multiple individual models, rather than two models overall.

Suggest 3 more models that examine meeting individual guidelines with obesity, one for the physical activity guideline, sleep guideline, and screen-time guideline. So, the meeting the physical activity guideline would be meeting physical activity guideline vs not meeting physical activity guideline with obesity. This may be more meaningful and comparable to other studies.

Malaysia’s income status should be featured in the introduction when comparing to other countries, and earlier in the paragraph when comparing to high income countries (Lines 281-293). It seems the authors did not compare to any reports of middle income countries, or explicitly state which countries would be comparable.

Line 312-313. Did SEANUTS use all the devices included within the current study? The difference may be including multiple devices included in the current questionnaire. The results still point to TV rather than non-TV devices, which should be considered in discussion.

Suggest rephrase 323-324 or delete as the sentence prior explains the same sentiment.

Line 337-339: The Mindell study also used a younger age range 3-5, which may explain differences, especially nap.

It is unclear the purpose of lines 352-363. Clarify if these are future directions for future research, or implications for practitioners. Please cite peer-reviewed literature to support these statements.

Reviewer 2 Report

Thank you for addressing my previous commentary systematically. I am satisfied that you have amended the manuscript accordingly. 

There are a few minor grammar/language issues to check. 

Author Response

This manuscript is a resubmission of an earlier submission. The following is a list of the peer review reports and author responses from that submission.

Round 1

Reviewer 1 Report

Thank you for the opportunity to review this paper. The current project aimed to describe the prevalence of the 24-Hour Movement Guidelines amongst a Malay population which has not yet been explored. My comments below are for the authors.

Major

-abstract should reflect which guidelines are used, i.e. the 4-5 y one and 6y one, which are different. Even a comment on age specific guidelines should be appropriate.

-include hours of MVPA also in abstract, rather than TPA

-person first language, have obesity rather than be obese.

-Provide detail on thinness and stunted for preschoolers in this area.

-Provide estimates of screen-time per device, possibly in one of the tables. Similar comment for nap-time and overnight sleep.

-Remove unpublished results and find literature to support claims on parenting.

Minor:

-Intro line 45-46, could you expand upon why there may not be associations in the 2-5 year old age range?

-Intro Line 50-51, None of these citations include original articles of US preschool 24-hour movement guidelines. Suggest Kracht et al., 2019 JPAH.

-Why were Malay, Chinese, and Indian requirements?

-Could you provide a quick overview of the kindergarten structure in this area? Is 4 years of age when most children in this country attend kindergarten? It is different in other countries.

-Provide more explanation on screen-time measure. Did the parents report separately for weekday and weekend? Were the parents asked to write in answers, or given categories choices?

-Provide more explanation on the sleep measure. Was the questionnaire for sleep validated? Were the parents reporting separately for naptime and overnight sleep? Were categorical options given, or did the parent write in the answer?

-Provide explanation of sex-stratified models.

-did you examine differences between 4 year olds and 5-6 year olds?

-Suggest adjusting for household income and/or maternal education within models.

-Lines 143-144, provide standard deviation with means.

-Line 146-147. Provide reference for stunted and how this was measured in methods. This should also be prefaced within the introduction if this is more common within this region.

-Provide estimates of screen-time per device, possibly in one of the tables. This may help with context of screen-time, and where the excess is coming from. Similar comment for nap-time and overnight sleep.

-is Malaysia a high-income country, or middle-income country? It would be appropriate to indicate it’s country status for these comparisons to namely high income countries.

-Suggest deleting sentence in lines in lines 248-250. The following sentence supports your claim.

-Paragraph 256-270 needs to be revisited, specifically 263-270. Authors should address the context of sleep, naps and overnight sleep, to examine these differences across countries. Suggest deleting lines 263-270, and adding moving up the next paragraph.

-Delete lines 278-280 for unpublished results or find other literature to support these conclusions.

-Line 284: change “24hr-movement” to 24-hour movement

Reviewer 2 Report

Thank you for submitting the article “Association between compliance with movement behaviour guidelines and obesity among Malaysian pre-schoolers” and interesting read.  Please find constructive commentary on this article:

The introduction is well written

After the study aims please insert the specific hypothesis being tested,

The methods are generally well written, expect please add details on procedure, who and how were participants issued with the belts, how were the parents provided with instructions, who conducted the height measurements and were they trained appropriately, etc.

From results:

Moreover note statistics here 15.3% of children, though this was higher in boys at 18% overweight or obese

Discussion-

Please refer back to the study hypothesis in the discussion as supported or not.

It is noteworthy that in table 2, children in the overweight category had the highlight minutes of PA, lowest screen time, and highest mean sleep duration, these figures drop significantly entering the obese category specifically. It is worth reflecting on this variation.

Line 206, remove the word Surprisingly.  Surprisingly to whom in what context and why, just removed.

about 15.3% of children were overweight and obese; while 4.7% and 7.4% of children were thin and stunted, respectively.- comment for the discussion, how do these statistic compare with national average (expected for cohort).

“The present study shows a very small percentage (6.5%) of preschoolers who met all three guidelines” link this finding back to prevalence of overweight  or obesity within the sample to demonstrate relationship.

SEANUTS Malaysia study (3 hours versus 1.5 hours) conducted a decade ago [37].. is this not expected as in the last decade the availability of smartphone, tablets, on-stream films has become more prevalence.

“The low compliance of screen time guidelines might possibly be due to the absence of rules imposed for screen activities at home.” Though may indicate that the screen time guidelines need further roll-out, promotion and education?

“Moreover, an increase of every minute of MVPA will also lead to the reduc- tion of half an hour of sleep time (results not shown)”  result not shown, why not, and then why comment on that here. Please expand this and make clear in the paper.  This is an interesting comment which needs evidence to back up and explain please.

There are a number of instances where “results not published”, “unpublished results”, “unpublished interviews results” are referred to in the discussion. This should be avoided.  If quantitative results are to be referred to in the discussion and enhance the claims of the paper they must be reference in the results section, and then discussed as appropriate.

There is no mention of a qualitative study and yet unpublished interview results are mentioned, this would be OK, if put in correct context.

The implications for practice (public health) – obesity prevention strategy and the implications for future research should be acknowledged clearly in the discussion.

References some formatting issues, e.g. italics/bold